# Random Copolymers of Lysine and Isoleucine for Efficient mRNA Delivery

**DOI:** 10.3390/ijms23105363

**Published:** 2022-05-11

**Authors:** Iuliia Pilipenko, Olga Korovkina, Nina Gubina, Viktoria Ekimova, Anastasia Ishutinova, Evgenia Korzhikova-Vlakh, Tatiana Tennikova, Viktor Korzhikov-Vlakh

**Affiliations:** 1Institute of Chemistry, St. Petersburg State University, Universitetskii pr. 26, Peterhof, 198504 St. Petersburg, Russia; pilipenko@scamt-itmo.ru (I.P.); olya_osipova_06_01@mail.ru (O.K.); nina.gubina2000@gmail.com (N.G.); vlakh@mail.ru (E.K.-V.); t.tennikova@spbu.ru (T.T.); 2International Laboratory “Solution Chemistry of Advanced Materials and Technologies”, ITMO University, Lomonosova St. 9, 191002 St. Petersburg, Russia; 3CJSC Biocad, ul. Svyazi., 34-A, Strelna, 198515 St. Petersburg, Russia; ekimova@biocad.ru (V.E.); ishutinova@biocad.ru (A.I.); 4Institute of Macromolecular Compounds, Russian Academy of Sciences, Bolshoy pr. 31, 199004 St. Petersburg, Russia

**Keywords:** mRNA delivery, random amphiphilic copolymers, cationic polypeptides, polyplexes, transfection

## Abstract

Messenger RNA (mRNA) is currently of great interest as a new category of therapeutic agent, which could be used for prevention or treatment of various diseases. For this mRNA requires effective delivery systems that will protect it from degradation, as well as allow cellular uptake and mRNA release. Random poly(lysine-co-isoleucine) polypeptides were synthesized and investigated as possible carriers for mRNA delivery. The polypeptides obtained under lysine:isoleucine monomer ratio equal to 80/20 were shown to give polyplexes with smaller size, positive ζ-potential and more than 90% encapsulation efficacy. The phase inversion method was proposed as best way for encapsulation of mRNA into polyplexes, which are based on obtained amphiphilic copolymers. These copolymers showed efficacy in protection of bound mRNA towards ribonuclease and lower toxicity as compared to lysine homopolymer. The poly(lysine-co-isoleucine) polypeptides showed greater than poly(ethyleneimine) efficacy as vectors for transfection of cells with green fluorescent protein and firefly luciferase encoding mRNAs. This allows us to consider obtained copolymers as promising candidates for mRNA delivery applications.

## 1. Introduction

Intracellular delivery of different genetic constructions in order to acquire the positive medical effect of such treatment represents emerging problem of modern biomedical science [1]. It is of common knowledge, that many approaches for treatment of different severe diseases require control over protein expression. The first type of genetic drugs was plasmid DNA (pDNA), which allowed inducing the cellular production of necessary protein [2]. However, pDNA effective action implies penetration to nucleus, which is quite challenging from delivery point of view. In this regard, application of messenger RNA (mRNA), which acts in cytoplasm, is more perspective [3]. Moreover, mRNA does not integrate to the genome of the cells, which benefits from potential carcinogenicity point of view [4].

In recent decades, the clinical role of mRNA-based therapeutics is constantly growing, mainly due to its great potential as vaccine [5]. mRNA vaccines played a crucial role in struggling COVID-19 pandemic [6]. However, mRNA vaccination is not only perspective for prevention of severe viral infections, but also possess great potential for immunotherapy of tumors [4,7,8]. The latter approach requires the transfection of antigen-presenting cells with mRNA, which encodes tumor-associated antigen [9] or chimeric antigen receptors (CARs) [10].

Despite some obvious clinical success, the development of mRNA delivery systems represents an urgent research problem. Although viral vectors are highly effective, they still have a number of drawbacks, such as potential to trigger immunogenic responses and transgene mis-insertion risks [4,11,12].

There are three major types of nanoplatforms applied for non-viral intracellular delivery of genetic constructions: lipoplexes, lipid nanoparticles and polyplexes [13]. First ones are composed of positively charged lipids, which form self-organized spherical vesicles due to presence of distinct hydrophilic and hydrophobic parts within their molecules, and could encapsulate nucleic acids (NA) into their inner aqueous phase [14]. Lipoplexes represent quite efficient class of transfection agents, which efficiently penetrate cellular interior and escape from endosomes [15,16,17]. Despite good efficacy in vitro, lipoplexes possess some limitations for in vivo applications, such as uncertain stability in serum [18] and toxicity [19].

These features of lipoplexes have turned the interest to lipid nanoparticles (LNPs) as NA delivery vehicles [20,21]. The morphology of LNPs differs from traditional liposome bilayer, and characterized by inverted micelle formed by cationic/ionizable lipids around the encapsulated NA molecules [22]. LNPs are more stable and quite versatile systems, which could serve as efficient NA delivery nanoplatforms in different applications [23]. However, further development of LNPs involves the possibility of combining LNPs with polycations in order to increase their intracellular penetration, endosomal escape and transfection efficacy [21].

Polycations represent an important type of non-viral vectors for delivery of genetic constructions [24,25]. During interaction of polycations with NA the condensation of NA into so called polyplexes (or polyion complexes) occurs [26]. This process is quite simple and reproducible, which makes it very attractive for future clinical applications. The obtained polyplexes are nanoparticles with high density leading for easy cell internalization and enhanced protection from enzymatic degradation [25]. The study of polycations application for stabilization of NA both in vitro and in vivo, as well as for promotion of nucleic acids intracellular penetration is vast and emerging. However, clinical application of polycations is still limited, mainly due to the low efficacy of such vectors. In order to give polycations a chance to serve as effective NA delivery vectors the study of different structural peculiarities of such polycations on the efficacy of NA binding, stabilization and transfection is of great importance.

There are three different physicochemical characteristics, which affect NA binding and transfection: positive charge, hydrogen bonding and hydrophobic interactions. While the first one is responsible for binding of NA, the second one is responsible for compactization of interpolyelectrolyte complexes (IPECs) into dense nanoparticles [27]. Hydrophobization of polycations usually benefits their ability for effective transfection and lowering toxicity [28,29,30]. The reason for this is the reduction of charge density by introduction of hydrophobic units, formation of stable and compact nanostructures, which better penetrate the cells. Hydrophobic modification of polycations is also believed to be responsible for better interaction with cell membranes and endosomolytic properties. In addition, hydrophobic interactions within the polyplex could cause the destabilization of polyplex, leading to better intracellular release of loaded NA [27,31].

Among different polycations many studies have been devoted to the application of poly(L-lysine) (PLys). PLys could be easily obtained by ring-opening polymerization of corresponding N-carboxyanhydride, and resulting polymer contains primary ε-amino groups in the side chains of each monomer unit, which provide great positive charge density. This allows an efficient binding of DNA or RNA molecules at physiological pHs. Other advantageous features of PLys are its biodegradability and efficient intracellular penetration. However, there are several drawbacks accompanying PLys application. These are toxicity for molecules with molecular weight (MW) above 30,000 and poor efficacy of transfection, which could be caused by too strong binding of NA with polycation.

Different research groups were focused on solving PLys poor transfection efficacy by chemical modification of PLys. Hydrophobization of PLys was performed in several studies and allowed to reach better results on transfection. Such hydrophobization could be performed by polymeranalogous reaction of prepared polymer, or by addition of hydrophobic monomer during polymer synthesis [32,33]. As example of the former approach, low and high MW (4000 and 25,000) PLys were modified by several endogenous lipids to substitute 10% of ε-NH2 groups of the polymer [34]. The transfection efficacy correlated with the degree of substitution, and overall, the lipid-modified high MW PLys was found to be about 20–25% more effective than unmodified PLys. Hydrophobized PLys modified during polymerization stage are generally represented by block copolymers. Self-assembled PEG-b-PLys-b-PLeu [35] micelles and PLys-b-PLeu with different length of Leu blocks [36] were obtained and studied as gene delivery vehicles. It was shown by the authors of mentioned studies that increasing of hydrophobic block length benefits the transfection efficacy.

Despite some number of studies devoted to the application of amphiphilic Lys-containing block-copolymers, there is an unexplored possibility of synthesis and application of random copolymers of lysine with other hydrophobic amino acids, such as leucine (Leu) or isoleucine (Ile), in NA delivery. Random copolymers of poly(amino acids) were described earlier [33,37] and showed different properties as compared to block-copolymers. One of the peculiarities of random copolymers is their flexibility, which could greatly affect NA binding and transfection. PLys is known to form α-helix [38], while Ile sequences in proteins are known to participate in the formation of β-sheets [39,40] (Figure 1). These structures are more or less rigid, which should affect the formation of polyplexes and intracellular release of entrapped genetic construction. Copolymerization of Lys and Ile into random copolymer should lead to flexible polymer chain with reduced positive charge as compared to pure PLys. At the same time, the presence of Ile should benefit penetration through cytoplasmic membrane and escape from early endosomes via hydrophobic interactions with phospholipids. Furthermore, decreasing of the charge density of polycation by introduction of hydrophobic units into should facilitate the release of NA from IPECs (Figure 1).

In this study, we have obtained random copolymers of lysine and isoleucine (P(Lys-co-Ile)) and assessed their properties in polyplexes with mRNA and pDNA. In addition, the study of their cytotoxicity and transfection efficacy were performed and discussed.

## 2. Results and Discussion

### 2.1. Synthesis of P(Lys-co-Ile) and Preparation of Polymer Particles

Amphiphilic cationic polypeptides P(Lys-co-Ile) were obtained by ring-opening polymerization of corresponding NCAs (Figure 2). The formation of copolymers was monitored by simultaneous appearance of aromatic signals of Z-protection at 7.1–7.4 ppm and aliphatic signals of isoleucine residue at 0.5–1.5 ppm in ^1^H NMR spectra. The removal of Z-protection groups was carried out by treatment with 9% trifluoromethanesulfonic acid (TFMSA) in trifluoroacetic acid (TFA) was detected by ^1^H NMR via disappearance of aromatic signals at 7.1–7.4 ppm.

PLys was synthesized under the same conditions and used as a benchmark for comparison. The molecular weight of polymers was controlled by applying different monomer to initiator molar ratios. As it was expected, the increasing of initiator content in the polymerization mixture has led to the decrease in polymeric product molecular weight and growth of dispersity. The comparison of polymerization of just Lys NCA with this of Lys and Ile NCAs mixture showed that addition of co-monomer decreases the molecular weight of the polymer (Table 1 and Appendix A). The amino acid composition of copolymers was determined via quantitative HPLC analysis of amino acids obtained after total acidic hydrolysis of copolymers (Appendix A). It was found that using of the Lys/Ile initial ratio equal to 80/20 (mol/mol) resulted in the formation of copolymers consisting of 85–87 mol% of Lys and 13–15 mol% of Ile (Table 1). This is caused by greater activity of Lys NCA in ring-opening polymerization as compared to Ile NCA.

It was shown that, in contrast with the homopolymer—PLys, amphiphilic P(Lys-co-Ile) copolymers could self-assemble into particles of different size depending on the copolymer molecular weight (Appendix A and Table 1). The decrease in polymer MW leads to the formation of more compact particles. All obtained particles possess prominent positive ζ-potential. This proves the capacity of these particles towards binding of nucleic acids.

The variation of monomers ratio in the polymerization mixture allowed the obtaining of copolymers with different contents of charged and hydrophobic parts (Appendix A and Table 1). The selection of the best copolymer composition was based on the analysis of particles size and particle size distribution via dynamic light scattering (Table 1), as well as on the ability of copolymer to effectively bind mRNA. P(Lys-co-Ile) copolymers with Lys/Ile molar ratios in polymerization mixture equal 60/40 and 70/30 gave quite large particles and poorly bind mRNA (see Appendix A). It should be noted that particles’ size is very important for intracellular penetration. Microparticles are known to less efficiently penetrate normal somatic cells as compared to nanoparticles. Thus, particles formed from Lys and Ile copolymer with a composition of 80/20 were selected as the most suitable for further work due to their smallest size and best mRNA binding efficacy (Appendix A).

The stability of obtained polypeptide particles was tested, and results of this test clearly showed that particles are quite stable for two weeks both in PBS, pH 7.4, and in FBS-free Opti MEM cell culture medium (Appendix A).

### 2.2. Effect of mRNA Binding on Particles Characteristics

One of the most important features of polycations designed for intracellular delivery of nucleic acids is their ability to bind nucleic acids (NA). The aim of this stage of research was to find the way of encapsulation, which will entrap the nucleic acid inside the particles and allow its intracellular penetration and transfection. Thus, we have tested the obtained polymers (P(Lys-co-Ile), 80/20) towards their capability to form polyplexes. We have tested three approaches to encapsulate firefly luciferase encoding messenger mRNA (fLuc-mRNA, further designated as just mRNA). These approaches are: (1) simple mixing of mRNA with polycation, (2) ultrasonication and (3) phase inversion. By applying these protocols, we were strived to obtain smallest particles with positively charged surface. The latter served as a confirmation of mRNA entrapment inside the particles.

It should be noted that in all cases mRNA was initially condensed by addition of Ca^2+^ ions. By using calcium complexation, the particle size and mRNA complexation were significantly improved. Notable, that no harmful effects of such complexation on cellular calcium homeostasis [41] were observed due to the low concentration of calcium ions used. According to the first approach polyplexes were formed by simple addition of polycation to condensed mRNA. Addition of mRNA at optimized N/P ratios leads to formation of more compact particles (Figure 3A). However, the analysis of ζ-potentials for formed polyplexes revealed negative charge for the smaller nanoparticles (Figure 3B). This could be explained by attaching of mRNA/Ca^2+^ to the surface of P(Lys-co-Ile) particles, which are quite flexible, but compact and stabilized by hydrophobic interactions. Thus, mRNA molecules could not be considered as encapsulated into polyplexes, but simply spun on polycation particles surface. Such location of mRNA molecules could lead to their rapid destruction by enzymes [42,43]. It should be noted that simple mixing of mRNA and polypeptide resulted in two modes on the particles size distribution curve obtained by DLS (see Appendix A in Appendix A), which is also the negative side of this method of obtaining polyplexes.

Under the third approach we have applied 30 s ultrasound (Sonopuls, 20% of power) on the stage of mRNA/Ca^2+^ mixing with cationic polypeptides to destabilize P(Lys-co-Ile) particles structure and allow encapsulation of mRNA molecules inside the polyplex structure. This method allowed to obtain small particles (Figure 4A) with constantly positive ζ-potential values (Figure 4B). Thus, it appears that ultrasonication leads to penetration of compacted mRNA/Ca^2+^ molecules into P(Lys-co-Ile) particles.

Moreover, it was observed that formation of polyplexes by ultrasonic processing results in formation of polyplexes with unimodal size distribution as compared to those obtained by just mixing (see Appendix A in Supporting information). Thus, ultrasonication could be considered as method of choice for obtaining of polyplexes as based on amphiphilic random copolymers such as P(Lys-co-Ile) or similar.

Despite the satisfactory characteristics of the particles obtained by ultrasonic processing, we are aware of the fact that the necessity of ultrasound treatment decreases the versatility of polycations application, since this equipment is not available in some laboratories and medical centers. For this reason, we have focused our efforts on developing a method, which allows formation of fine particles with structure needed without additional ultrasonication.

Encapsulation of mRNA into P(Lys-co-Ile) was possible by applying third approach, namely, phase inversion method. For that, polypeptides were dissolved in a 70% ethanol-water mixture at concentration of 1 mg/mL. The predetermined volume of polymer solution was mixed with mRNA/Ca^2+^ and incubated during 30 min. The final particles with encapsulated mRNA were formed during removal of ethanol, which was performed by sequential addition of 0.01 M phosphate buffer saline, pH 7.4 (PBS 7.4) and ultracentrifugation (Figure 5A).

One can observe that the phase inversion procedure allows the formation of particles with D_H_ and ζ-potential values (Figure 5B,C) similar to those obtained with application of ultrasound. However, in the case of phase inversion approach P(Lys-co-Ile) samples with smallest molecular weight gave particles with smaller size and higher ζ-potential. No visible effect of applying different ethanol/water ratios on characteristics of the particles was detected. Importantly, that ζ-potential values of obtained particles were positive. This indicates the location of Mrna inside the polyplexes and proves efficient encapsulation of Mrna.

The morphology of particles obtained by ultrasonication and phase inversion method were studied by TEM (Figure 6). It was observed that particles obtained by phase inversion were spherical and less aggregated structures as compared to polyplexes formed during ultrasound treatment.

### 2.3. Mrna Encapsulation Efficacy

The successful application of gene delivery systems implies efficient loading of nucleic acid into the polyplex. Thus, we have quantified the ability of obtained polycations to encapsulate the target Mrna. For this purpose, we have applied RiboGreen fluorometric assay, which allows to detect RNA concentrations in a very sensitive manner [44]. The determination of encapsulation efficiency via direct fluorescence detection was based on the experimental fact that only free Mrna, but not encapsulated one, will be stained with the dye. One can observe that efficacy of Mrna binding was nearly the same in the case of applying ultrasound treatment and phase inversion methods (Figure 7A). For this reason, we have applied more simple and versatile phase inversion method in all further experiments. Considering that P(Lys-co-Ile)-3 sample allowed to obtain the smallest particles (Figure 5B), this copolymer was used in most of further experiments.

Of particular interest are results obtained with specially synthesized PLys-PEG copolymer. The idea of synthesizing this polymer was to prolong the circulation of polyplex in the body due to well-known “stealth” effect [19]. However, we observed the significant drop in the efficiency of mRNA encapsulation for PLys-PEG as compared to PLys and P(Lys-co-Ile) copolymers (Figure 7A). Thus, we decided to exclude PEGylation of polycations from our further experiments.

Encapsulation of mRNA into polypeptide-based polyplexes obtained via phase inversion was also assessed by electrophoresis in agarose gel. The results showed that the mobility of mRNA was completely retarded at chosen N/P, which proves the high efficiency of this nucleic acid binding within obtained polyplexes (Figure 7B).

Synthesized P(Lys-co-Ile) polypeptides were also able to efficiently bind not only mRNA, but also DNA (Appendix A, Appendix A).

### 2.4. mRNA Stabilization towards Enzymatic Degradation

One of the important features of polyplexes is stabilization of NA towards enzymatic degradation. Here, we have studied the effect of mRNA encapsulation into P(Lys-co-Ile) and PLys polyplexes on its resistance to the action of RNase. For that, pure mRNA and its complexes with cationic polypeptides (N/P = 2,4 and 8) were coincubated with enzyme. The degradation and stability of mRNA were detected by 1% agarose denaturing gel electrophoresis (Figure 8). Comparison of lanes 1 and 4 clearly shows that unprotected at low N/P 2 mRNA can be degraded by RNase. Complexation of mRNA within polyplexes based on PLys and P(Lys-co-Ile) at N/P 4 and N/P 8 resulted in the stabilization of mRNA towards degradation by RNase (Lanes 2,3 and 5,6, respectively).

In addition, the ability of polymers to stabilize the DNA was proved (Appendix A, Appendix A).

### 2.5. Cytotoxicity of Polymers and Their Polyplexes with mRNA

An important challenge in creating effective mRNA delivery systems is to reduce their toxicity. It is well known, that most polycations, which can serve as effective gene carriers, namely PEI, PLys, PAMAM etc., are quite cytotoxic. This toxicity is provoked by positive charge of macromolecules, which cause disruption of intercellular contacts and contacts with the intercellular matrix, as well as cell lysis [45]. Thus, diminishing of positive charge density is important for reduction of cytotoxic effect. In P(Lys-co-Ile) copolymers the positive charge is reduced due to presence of hydrophobic amino acid.

In this study human epithelial cells were used to compare the cytotoxicity of PLys and P(Lys-co-Ile) copolymers (Figure 9). Epithelial cells are important targets for transfection, because their layers operate in the organism in multiple ways, all of which are crucial epithelial transport, secretion of proteins, spatial ion buffering, phagocytosis and immune regulation [46,47]. One can observe that PLys itself is more toxic than P(Lys-co-Ile). However, both polymers possess certain toxicity to the cells. The addition of mRNA resulted in increasing of cells viability, both in the case of Lys homopolymer and its copolymer with Ile. The increasing of polypeptide content within the polyplex resulted in growth of its toxicity for the cells under study. It appears that polyplexes are slightly more toxic for ARPE-19 cells, than for HCE ones. It could be also seen from the obtained results that PLys based polyplexes began to be toxic at N/P 8, while those base on P(Lys-co-Ile) show toxicity only at N/P above 10. Thus P(Lys-co-Ile) copolymers could be considered as fewer toxic alternatives to PLys.

### 2.6. Particles Intracellular Penetration

The ability of the polyplexes for effective intracellular penetration is an important prerequisite for their application as transfection agent. In contrast with transfection of cells with pDNA, which should be delivered into nucleus, mRNA effectively initiates the expression of desired protein after delivery to cytosol. Here we have tested the ability of polyplexes based on P(Lys-co-Ile) and EGFP-mRNA to penetrate the cytosol. In order to detect the intracellular penetration of polyplexes Cy3 labeled oligonucleotide duplex, namely oligo-dT-dA (Cy3-oligo-dT-dA), was included into composition of these polyplexes. The hydrodynamic diameter of P(Lys-co-Ile)/EGFP-mRNA/Cy3-oligo-dT-dA particles (162 ± 34 nm) was similar to those of just P(Lys-co-Ile)/EGFP-mRNA (155 ± 41 nm). Fluorescent labeling of oligonucleotide allowed us to detect its penetration to the ARPE-19 cells via fluorescent microscopy (Figure 10A). The presented images clearly show the concentration of the labeled oligonucleotide in the cytosol of the cells near the DAPI stained nucleus.

The quantitative measuring of cells intracellular penetration reveals the greater potential of P(Lys-co-Ile) towards intracellular penetration (Figure 10B), which could be explained by the importance of hydrophobic interactions with lipid bilayer for such process.

### 2.7. Transfection Studies

The target property of the created polypeptides is the ability to transfect cells. Usually, the transfection efficacy of PLys itself is quite low. In this study, we performed a comparative transfection study using PLys, P(Lys-co-Ile) copolymer and branched PEI with *M_n_* of 25,000 (bPEI 25k) as control.

First, we have tested the obtained polymers as agents for transfection of ARPE-19 and HCE cells with fLuc-pDNA-Ca^2+^. In this experiment, we have tested different N/P ratios to find the optimized ones, which show better transfections efficacy as related to transfection provided by bPEI 25k (see Appendix A in Appendix A). It was observed that transfection of pDNA with application of both PLys and P(Lys-co-Ile) is less effective than that provided by bPEI 25k. At the same time P(Lys-co-Ile) showed better efficacy towards transfection of pDNA than just PLys. One can observe that most efficient transfection with application of P(Lys-co-Ile) was observed at N/P 4.

The transfection efficacy of K562 immortalized myelogenous leukemia cell line with EGFP-mRNA was tested via flow cytometry (Appendix A). In this experiment naked EGFP-mRNA and this complexed with Lipofectamine 2000 transfection agent were used as controls, while P(Lys-co-Ile) polyplex with EGFP/mRNA/Ca^2+^ was the sample under study. One can observe that transfection with lipofectamine is the most after 24 h, while for P(Lys-co-Ile) the maximum transfection was observed only after 48 h. This observation could be explained by greater size of polyplex particles than those of lipoplexes. It is known that smaller particles better penetrate cellular interior and better act as transfection agents [48,49,50].

The results of ARPE-19 and HCE cells transfection by EGFP-mRNA and fLuc-mRNA showed more efficient target protein expression as compared to transfection of cells with pDNA (Figure 11).

It could be concluded that synthesized copolymers do not provide effective delivery of pDNA to the cell nucleus but can quite efficiently deliver mRNA to cytoplasm. Interestingly, that obtained images (Figure 11A) show increased number of ARPE-19 cells with GFP fluorescence especially in the case of amphiphilic random copolymer of Lys and Ile, rather than in the case of PLys. P(Lys-co-Ile) vector also showed considerably better efficiency of transfection of both ARPE-19 (Figure 11B) and HCE (Figure 11C) cells with fLuc-mRNA as compared with that in the case of bPEI 25k and PLys. Thus, it is obvious that introduction of hydrophobic amino acid in polycation structure enhances the efficacy of mRNA delivery to the cell cytoplasm. This effect could be explained by better interaction of P(Lys-co-Ile) polyplexes with cell membranes, as well as by more easy intracellular release of mRNA. The latter is favored by flexibility of polycation chain as well as diminishing of positive charge density due to introduction of Ile into PLys structure.

It is important to note that the best transfection efficacy was observed with application of P(Lys-co-Ile)-3 (Table 1). This gives evidence for more efficient transfection of the cells under study with application of cationic peptides with low molecular weight. The possible explanation for such an effect is that macromolecules with low molecular weight are more easily release the encapsulated mRNA than polypeptides with higher degree of polymerization. Moreover, polypeptides with lower molecular weight are less toxic, which results in better survival of transfected cells.

## 3. Materials and Methods

### 3.1. Materials

ε-Z-L-lysine (Lys(Z)), L-isoleucine (Ile), α-pinene, triphosgene, trifluoroacetic acid (TFA), n-butylamine, trifluoromethanesulfonic acid (TFMSA), dimethyl sulfoxide-d6 (DMSO-d6, 99.8%), heparin sodium salt (Mw 8000–12,000), Ribonuclease A (RNase) were obtained from Sigma–Aldrich (Darmstadt, Germany) and used without additional purification. All organic solvents, i.e., N,N-dimethylformamide (DMF), dimethyl sulfoxide (DMSO) 1,4-dioxane, petroleum ether, ethyl acetate and some others were purchased from Vecton Ltd. (St. Petersburg, Russia), and purified before use according to standard protocols. For purification of synthesized polymers, Spectra/Pore^®^ dialysis bags (MWCO:1000 and 10,000, Rancho Dominguez, CA, USA) were used. Amicon Ultra filter tubes with 10,000 and 50,000 MWCO (0.5 mL) were purchased from Merck (Darmstadt, Germany). Cy3-NHS fluorescent dye was purchased from Lumiprobe (Moscow, Russia). Ethidium bromide (EtBr) and agarose were purchased from Molecular Probes (Eugene, OR, USA).

SEC column calibration was performed with the use of poly(methyl methacrylate) (PMMA) standards (Mw 17,000–250,000; Đ ≤ 1.14) purchased from Supelco (Bellefonte, PA, USA). Spin columns (molecular weight cut-off (MWCO) 3000; VivaScience, Sartorius Group, Göttingen, Germany) and Eppendorf tubes with filter (30,000 MWCO, Amicon Ultra 0.5 mL, Merck, Darmstadt, Germany) were used for dialysis, IPECs purification, and separation.

Cy3-labeled 23-base pairs duplex of oligothymidine and oligoadenine (oligo-dT-dA) was purchased from Biobeagle (St. Petersburg, Russia). Firefly luciferase encoding mRNA was prepared as described below. Plasmids encoding luciferase (pCLuc4) and GFP (pEGFP-C2) were a kind gifts of Dr. Marika Ruponen (School of Pharmacy, University of Eastern Finland, Kuopio, Finland). GFP encoding messenger RNA (EGFP-mRNA) was obtained from IBA (Göttingen, Germany). Lipofectamine 2000 Transfection agent (Invitrogen, Carlsbad, CA, USA).

Branched poly(ethylene imine) (bPEI 25k; M_w_ = 25,000 and M_n_ = 10,000 according to GPC; Sigma-Aldrich, St Louis, MO, USA) was used as control in transfection studies.

The plastic for cell cultivation was obtained from Sigma-Aldrich (Darmstadt, Germany) and Sarstedt AG and Co (Nümbrecht, Germany). Transfection analysis was performed with Gaussia Luciferase kit (Thermo Fisher Scientific, Roskilde, Denmark).

All other materials are described further upon their appearance in the text.

### 3.2. Instruments

The magnetic stirrer MR Hei-Mix S (Heidolph, Schwabach, Germany), Schlenk reaction tubes with rubber septum (Aldrich, Munich, Germany) and rotary evaporator Hei-VAP Precision ML/G3B (Schwabach, Heidolph, Germany) were used for polymer synthesis. ^1^H NMR spectroscopic data were recorded with equipment of Magnetic Resonance Research Centre of St. Petersburg State University: Bruker Avance spectrometer (400.13 MHz for 1H and 100.61 MHz for 13C) in DMSO-d6 or in D2O. Shimadzu LC-20 Prominence system supplied with refractometric RID 10-A detector (Kyoto, Japan) and 7.5 × 300 mm Agilent PLgel MIXED-D column (Chrom Tech, Apple Valley, MN, USA) were applied for size exclusion chromatography (SEC) analysis.

An ultrasound homogenizer (Sonopuls HD2070, Bandelin, Berlin, Germany), Vortex (Thermo Fisher Scientific, Waltham, MA, USA) and TS-100C thermo-shaker BioSan (Riga, Latvia) were applied for polyplexes formation. The Vivaspin column ultra-filtration and particle separation were performed with Sigma 2–16 KL centrifuge (Sigma, Darmstadt, Germany).

A dynamic and electrophoretic light-scattering (DLS and ELS) instrument Zetasizer Nano ZS (Malvern, Enigma Business Park, United Kingdom) equipped with a He–Ne laser beam at λ = 633 nm was used for measurements of particle hydrodynamic radius, particle size distribution and surface charge. Nucleic acids were quantified spectrophotometrically with application of Nanodrop 2000c spectrophotometer (Thermo Fisher Scientific, Waltham, MA, USA). The photometrical detection of aldehyde groups was performed with UV–Vis-1800 Shimadzu (Kyoto, Japan). Microplate Spectrophotometer-Fluorometer Fluoroskan Ascent reader (Thermo Fisher Scientific, Waltham, MA, USA) was used for quantitative analysis. Nanoparticles morphology was analyzed in Centre for Molecular and Cell Technologies of St. Petersburg State University by using transmission electron microscope Zeiss Libra 200FE (Carl Zeiss, Oberkochen, Germany).

Agarose gel electrophoresis was performed with BlueMarine 200 system (Serva Electrophoresis GmbH, Heidelberg, Germany) and analyzed with SkyLight Table ECX-F20 (Vilber, Collégien, France).

PCR amplification was performed with equipment purchased in Biometra Thistle Scientific (Glasgow, UK).

The cell uptake efficiency was determined using CELENA S Digital Imaging System (Logos Biosystems, GE Healthcare, South Korea). Transfection efficiency was visualized by the Cytell Cell Imaging instrument (GE Healthcare, Washington, Issaquah, USA) and quantified with Microplate Spectrophotometer-Fluorometer Fluoroskan Ascent reader (Thermo Fisher Scientific, Waltham, MA, USA).

All other instruments are described further upon their appearance in the text.

### 3.3. Cells Culturing

Human retinal pigment epithelial (ARPE-19) and corneal epithelial (HCE) cells were obtained from American Type Culture Collection (ATCC, Manassas, VA, USA). Medium materials for cell culture were obtained as follows. ARPE-19 growth medium contained Dulbecco’s modified eagle medium (DMEM, Gibco Laboratories, Gaithersburg, MD, USA), fetal bovine serum (FBS, 10 vol%, Thermo Fischer Scientific, Roskilde, Denmark), L-glutamine (1 vol%), penicillin (1 vol%), streptomycin (1 vol%). 500 mL of HCE growth medium contained 402.5 mL of DMEM F12 (Gibco Laboratories, Gaithersburg, MD, USA), 75 mL of FBS (Thermo Fischer Scientific, Roskilde, Denmark), penicillin (1 vol%), streptomycin (1 vol%), L-Glutamine (5 mL), epidermal growth factor (EGF, 0.005 mg), insulin (2.5 mg), DMSO (2.5 mL), cholera toxin (0.05 mg), 1 M HEPES (7.5 mL, Thermo Fischer Scientific, Roskilde, Denmark).

### 3.4. Methods

#### 3.4.1. Synthesis and Characterization of Cationic Polypeptides

Polypeptides were synthesized by ring-opening polymerization (ROP) of α-amino acid N-carboxyanhydrides (NCA) as random structures (Figure 2). NCA monomers of Lys(Z) and Ile were prepared via the reaction of α-amino acids with triphosgene in anhydrous dioxane under argon atmosphere with addition of α-pinene [33]. Synthesized NCAs were purified by recrystallization from anhydrous ethyl acetate/n-hexane. Yields: Lys(Z) NCA—85%, Ile NCA—69%. The structure and purity of NCAs were confirmed by 1H NMR at 25 °C in CDCl3 (400 MHz Avance instrument, Bruker, Karlsruhe, Germany). Lys(Z) NCA: δ 7.43–7.28 (m, 5H), 6.97 (s, 1H), 5.12 (s, 2H), 4.97 (s, 1H), 4.32–4.23 (t, J = 5.2, 1H) (s, 1H), 3.29–3.14 (m, 2H), 2.03–1.90 (m, 1H), 1.90–1.75 (m, 1H), 1.73–1.28 (m, 4H); Ile NCA: 0.836 (t, 3H), 0.871 (d, 3H), 1.236 (dq, 2H), 1.941 (qtd,1H), 4.28 (d, 1H).

NCAs were polymerized to produce P(Lys(Z)-co-Ile) using n-hexylamine as initiator and the following molar ratio of monomers: [Lys(Z) NCA]/[Ile NCA] = 80/20. The NCAs to initiator molar ratios varied as 10, 20 and 50. Polymerization was initiated by dropwise addition of n-hexylamine to 4 wt% solution of NCAs in anhydrous 1,4-dioxane. The mixture was stirred under inert atmosphere at room temperature during 48 h. The product was precipitated with 50 volume excess of diethyl ether. The precipitate was dissolved in DMF and dialyzed against water for 36 h (MWCO 1000).

Molecular-weight characteristics (weight average and number average molecular weights, Mw and Mn, respectively, as well as dispersity, Đ) of polymers were evaluated by SEC with application of DMF with 0.1 M LiBr was used as a mobile phase. The analysis was performed at 40 °C under 1.0 mL/min of the mobile phase flow rate. SEC LC Solutions software (Shimadzu, Kyoto, Japan) was used for calculations of Mw, Mn and Đ regarding the calibration curve plotted for PMMA standards.

The amino acid compositions of obtained polypeptides were determined using 1H NMR at 25 °C in DMSO-d6 at 400 MHz. PLys, δ ppm: n-hexylamine—0.8 (CH_3_); lysine—4.1–4.3 (C**H**), 2.97 (NH-C**H**_2_), Z-protection—5.0 (O-C**H**_2_-C_6_H_5_), 7.1–7.4 (O-CH2-C6**H**5); P(Lys-co-Ile): δ ppm: n-hexylamine—0.9 (C**H**_3_), lysine—4.1–4.3 (C**H**), 3.0 (NH-C**H**_2_), Z-protection—5.0 (O-C**H**_2_-C_6_H_5_); 7.1–7.4 (O-CH_2_-C_6_**H**_5_); isoleucine—0.5–0.8 (C**H**_3_), 1.1–1.4 (C**H**_2_CH_3_), 1.5–1.7 (C**H**(CH_3_)(C_2_H_5_)), 3.4–3.6 ((CO)(NH)**CH**CH(CH3)(C_2_H_5_)).

The Z-protection groups were removed by 9% trifluoromethanesulfonic acid in trifluoroacetic acid (TFMSA/TFA, Sigma Aldrich) according to general procedure [27]. The peptide products were dissolved in DMF, dialyzed against water for 36 h (MWCO 1000) and lyophilized under vacuum. The removal of Z-protection was detected with application of ^1^H NMR in D_2_O by disappearance of aromatic signals at 7.1–7.4 ppm.

Composition of copolymers was determined via quantitative HPLC analysis of free amino acids obtained after total acidic hydrolysis of copolymers with the use of protocol developed earlier [33]. An example of chromatogram as well as details on chromatographic conditions and calibration plots can be found in Appendix A (Appendix A).

#### 3.4.2. Preparation and Characterization of Polypeptide Particles

Polypeptide nanoparticles suspensions in 0.01 M PBS at pH 7.4 were obtained by 30 s sonication treatment of polycations using 10% power of ultrasonic homogenizer with application of ultrasound homogenizer (70 W, acoustic energy—0.21 kJ).

Average hydrodynamic diameter of particles (D_H_) and polydispersity index (PDI) were measured at 25 °C by DLS at scattering angle of 173°. DLS experiments were performed in 0.01 M PBS, pH 7.4, whereas ζ-potential determination by ELS was carried out in deionized water. Stability of peptide nanoparticles colloidal suspensions were examined by DLS in PBS pH 7.4 and Opti MEM (FBS-free) for 14 days.

The morphology was investigated by transmission electron microscopy (TEM). The samples were prepared by dropping 3 μL of particles’ suspension (0.3 mg/mL) on copper grid (300 mesh) covered with carbon and formvar, and further staining with 3 w/w% uranyl acetate solution for 1 min. The grids were washed gently with pure water and dried for 30 min before the measurement.

#### 3.4.3. In Vitro Transcription of mRNA

The gen of interest (firefly luciferase) mRNA (fLuc mRNA) was cloned into a modified pSMART vector. The open reading frame of the gene was flanked by the Kozak sequence in the 5′UTR and the human a-globin sequence in the 3′UTR and poly A tail. The template for in vitro transcription was prepared by PCR (Biometra Thistle Scientific, Glasgow, UK) amplification of the gene. The required amount of plasmid was linearized by SpeI restriction enzyme.

mRNA was synthesized with the T7 RNA polymerase [51]. 500 μL of reaction contained: 50 ng/μL linearized plasmid (BIOCAD, Saint Petersburg, Russia), 2 mM Ribonucleotide Triphosphates (NEB, Ipswich, MA, USA), 1x Transcription buffer (40 mM TrisHCl pH 8, (Sigma Aldrich, St. Louis, MO, USA), 6 mM MgCl_2_ (Sigma Aldrich, St. Louis, MO, USA), 2 mM spermidine (Panreac, Barcelona, Spain), 5 mM DTT (Panreac, Barcelona, Spain), 1 unit/μL RNaseIn (Solar Bio, Beijing, China), 0.2 μg/μL T7 RNA polymerase (BIOCAD, Saint Petersburg, Russia). The reaction was incubated at 37 °C for 4 h. The product was treated with 25 units of DNase I (Neb, Ipswich, MA, USA) and purified by LiCl (Sigma Aldrich, St. Louis, MO, USA) precipitation. The capping reaction was performed by Vaccinia Capping System [52]. 1000 μL of reaction medium contained 1x capping buffer (50 mM Tris-HCl pH 8.5 mM KCl (Panreac, Barcelona, Spain), 1 mM MgCl_2_, 1 mM DTT), 0.5 mM Guanosine Triphosphate (Neb, Ipswich, MA, USA), 0.1 mM S-Adenosyl methionine (Neb, Ipswich, MA, USA), 1U/μL RNAseIn, 7.5 ng/μL Vaccinia Capping Enzyme (BIOCAD, Saint Petersburg, Russia), 500 μg of denaturated RNA. The reaction medium was incubated at 37 °C for 1 h. The final Luc mRNA was purified with silica columns [26] (Promega, Madison, WI, USA).

The 1 mg/mL stock solution of nuclease resistant Luc mRNA in 1 mM sodium citrate buffer was prepared and stored in the freezer. The stability of Luc mRNA was testified before application by 1% agarose denaturing gel electrophoresis (Appendix A, Appendix A).

#### 3.4.4. Condensation of mRNA with Calcium Chloride

mRNA was turned into condensed form by calcium chloride at 20:1 [nucleotide:Ca^2+^] molar ratios. Briefly, 10 µg of mRNA was mixed with 0.16 µg of CaCl_2_, the mixture volume was then adjusted to 100 µL using 1xTE buffer. Calcium chloride was overall used at a concentration of 1.4 × 10^−7^ M established as safe concentration for intra- and extracellular environment [41]. Hydrodynamic size and ζ-potential of prepared mRNA-Ca^2+^ complexes were 87 nm (PDI 0.22) and −32.9 mV, respectively.

#### 3.4.5. Encapsulation of mRNA

mRNA-Ca^2+^ complexes were then incorporated into positively charged polypeptides through polyelectrolyte complexation between phosphate groups and ε-amino groups of lysine with varying N/P ratio [nitrogen to phosphate] using three methods.

Method 1: the polymers (P(Lys-co-Ile), PLys) and NA solutions (1 mg/mL) were mixed at predetermined N/P ratios and vortexed at 1000 rpm, and 25 °C during 30 s. The formed suspensions were further incubated at 25 °C during 30 min for particles ageing.

Method 2: the dispersion of polymer particles in water or buffer solution (1 mg/mL) was prepared as in method 1, but then subjected to ultrasonication during 30 s and diluted to a desired concentration. After that, the condensed mRNA-Ca^2+^ complex was quickly added to the suspension under 1500 rpm vortex stirring during 2 min. The mixture was left for 30 min at room temperature for ageing.

Method 3: the polymers (P(Lys-co-Ile); PLys) were dissolved in a 70% ethanol-water mixture at concentration of 1 mg/mL. Particles were formed varying N/P ratio [nitrogen to phosphate]. For that, the appropriate volume of polymer solution was quickly mixed with mRNA-Ca^2+^ complexes (prepared in TE buffer) at 1500 rpm stirring, the reaction mixture was left for 30 min at room temperature and then washed with 0.01 M PBS pH 7.4 via ultracentrifugation (MWCO 10,000) to purify from ethanol.

The concentrations of mRNA were evaluated using RiboGreen fluorometric assay (Promega, Madison, WI, USA) [44]. Ribogreen dye was diluted 200 times with a TE buffer in a dark plastic tube and stirred. For the calibration curve RNA standard solution (Ribosomal RNA standard, 16S and 23S rRNA from E. Coli) was diluted with a TE buffer to prepare final concentrations from 20 to 1000 ng/mL. 100 µL of each RNA concentration was mixed with 100 µL of Ribogreen solution in 96-well plates, the samples were kept at room temperature for 2–5 min and then analyzed using fluorimetry at an excitation wavelength of 480 nm and an emission wavelength 520 nm. mRNA sample was diluted with a TE buffer to a total mRNA concentration of 2 µg/mL. The concentration of mRNA samples was determined according to the same procedure and calculated via standard RNA calibration curve.

Effective binding of mRNA within polypeptide particles prevents staining of this NA with RiboGreen. Thus, only free mRNA was stained. This allowed direct determination of mRNA encapsulation efficacy without the need for separation of particles and the dye. For that, 100 µL of each particle suspension was mixed with 100 µL of Ribogreen solution in 96-well plates, the samples were analyzed as described above. Encapsulation efficiency was calculated using the following equation:EE (%) = (m_1_ − m_2_)/m_1_ × 100(1)
where m_1_ is the initial input of mRNA, m_2_ is the amount of non-bound mRNA determined by its fluorescence with application of RiboGreen Assay.

#### 3.4.6. Agarose Gel Electrophoresis

To prepare the denaturing gel, 1 g of agarose was dissolved in 72 mL of water under heating, then 10 mL of 10X MOPS running buffer and 18 mL 37% formaldehyde (12.3 M) was added to the flask. The gel was poured using a well-comb and 1X MOPS was used as a running buffer. RNA samples were prepared as follows: 600 ng of free mRNA or particles, containing 600 ng RNA, were mixed with the equal volume of 1X Formaldehyde Loading Dye, containing ethidium bromide for visualization at a final concentration of 10 mg/mL. Samples were heated at 65–70 °C for 15 min and loaded to the gel. Electrophoresis was carried out at 6 V/cm until the bromophenol blue migrated at least ⅔ of the gel length and then the gels were imaged.

To study the enzymatic stabilization of mRNA by polyplexes RNase solution 3 µL (15 µg/mL) was added to the solution of pure mRNA 15 µL (150 ng) and 15 µL of polyplexes suspensions, containing the equivalent amount of mRNA. The reaction was stopped by addition of 2-fold wt/wt heparin towards RNase. To release mRNA from polyplexes before or after coincubation with RNase the 8-fold wt/wt excess of heparin towards polycation was added to polyplexes. Released mRNA was purified from heparin and polycation with application of Vivaspin columns (MWCO 100,000).

#### 3.4.7. Cytotoxicity

The cytotoxicity of polyplexes with fLuc-mRNA prepared with varying N/P ratio from 2:1 to 20:1 was studied using 3-(4,5-dimethylthiazol-2-yl)-2,5-diphenyltetrazolium bromide (MTT) reduction assay. Human retinal pigment epithelial (ARPE-19) and human corneal epithelial (HCE) cell lines were seeded into 96-well plates at a density of 20,000 and 23,000 cells/well. The cells were cultured in ARPE-19 and HCE cell culture media prepared according to the protocol (Experimental Section 2.2). After 24 h, 10 µL of nanoparticles were added in 190 µL of Dulbecco’s Modified Eagle Medium (DMEM) F-12 with 10% fetal bovine serum (FBS) to each well in triplicate. The cells were exposed to the nanoparticles for 24 and 72 h. Thereafter, the medium was replaced with 10 μL MTT solution (5 mg/mL in DMEM F-12; filtered through 0.45 μm) and 100 μL of fresh DMEM F-12. The cells were incubated for 2 h at 37 °C and 5% CO2. Then, 100 μL of lysis buffer with sodium dodecyl sulfate and dimethylformamide was added to each well and the cells were incubated for an additional 3 h at 37 °C and 5% CO2. The absorbance of the wells was measured at 570 nm using a microplate reader (Viktor 2, Perkin Elmer, Waltham, MA, USA). Polypeptide without mRNA (PP) was used as control. The negative controls were DMEM F-12 (untreated cells) and 10% PBS (pH 7.4) in DMEM F-12 (solvent control). The mean value of solvent control (corrected for blank value) was set as 100% cell viability. The relative cell viability was calculated as following:Cell viability (%) = (A_sample_ − A_blank_)/(A_control_ − A_blank_) × 100(2)

#### 3.4.8. Cell Uptake

Cell uptake of polypeptide-mRNA nanoparticles into ARPE-19 cell line was assessed using Cy3 labeled polypeptides. The cells were incubated with nanoparticles in serum-free medium for 4 h, then the medium was removed and cells were washed with 1 M NaCl in order to wash out not penetrated particles. Then, 100 μL DMEM-F12 containing 2 × FBS and 2 × penicillin–streptomycin was added for another 20 h of incubation. After 24 h the cells were fixed using exposure of 200 μL 3.7% of formaldehyde–methanol per well for 15 min at 37 °C, washed three times with PBS. Cell membranes were permeabilized with 0.2% Triton X-100 in PBS for 15 min. Staining of the cell nuclei was performed using DAPI for 30 min according to a manufacturer protocol. The cells were washed with PBS and then with distilled water for three times. The cell uptake efficiency was determined by analyzing the fluorescence intensity of Cy3-polypeptide nanoparticles (λ_ex_ = 532 nm, λ_em_ = 556 nm) using CELENA S Digital Imaging System (Logos Biosystems, GE Healthcare, Anyang, Kyonggi-do, South Korea).

#### 3.4.9. Transfection of Cells with Plasmid DNA

To visualize the efficiency of nanocarriers developed to deliver and release nucleic acids, the transfection of ARPE-19 was performed using polypeptide nanoparticles with encapsulated 100 ng of plasmid DNA (pEGFP-C2) at N/P 4:1. Transfection efficiency was visualized by the Cytell Cell Imaging instrument (GE Healthcare, Washington, Issaquah, WA, USA).

Quantitative analysis on transfection efficiency of ARPE-19 and HCE was assessed using polypeptide particles, containing mRNA or pDNA coding luciferase genes.

#### 3.4.10. Transfection of Cells with fLuc-mRNA

Cells (ARPE-19 and HCT) were seeded at a density of 50,000 cells per well in a 48-well plate. Next day, the cell culture medium was refreshed and 50 µL of nanoparticles, containing 1 µg of mRNA luc, with N/P ratio of 4:1 and 8:1 was added to each well. The cells were incubated with nanoparticles for 24 and 48 h separately. Substrate (One Glo Luciferase Assay System) was added directly to the cells in each well in a volume ratio of 1:1 (volume of cell liquid to substrate volume). Luminescence was measured after 10 min of incubation with substrate in a Black Corning plate with application of Microplate Spectrophotometer-Fluorometer Fluoroskan Ascent reader. The percentage of luminescence in the tested samples was normalized relative to the luminescence after transfection using bPEI-fLuc-mRNA, which was taken as 100%.

## 4. Conclusions

In this study, we have synthesized new polypeptides, namely, random copolymers of Lys and Ile. The best properties of polyplexes, such as size, ζ-potential and encapsulation efficacy were observed for polypeptides consisting of Lys and Ile in a ratio of 80/20 (mol%). Being amphiphilic polypeptides, P(Lys-co-Ile) could self-assemble into particles. Simple mixing of nucleic acids with such polymers in aqueous media results in spooling of nucleic acids over the polypeptide particles. In order to encapsulate nucleic acids into the inner part of polyplexes ultrasonication or phase inversion procedures could be applied. The latter one is more versatile, because it doesn’t require the ultrasound equipment, but only centrifuge, and results in spherical particles with smaller size.

The obtained P(Lys-co-Ile) polyplexes allowed effective protection of encapsulated mRNA towards RNAse. The toxicity of copolymers was significantly lower than that of just PLys, which is in good correlation with diminishing of charge density within obtained structures. The P(Lys-co-Ile) based polyplexes were found to effectively penetrate cells. These polymers also showed efficacy as vectors for transfection of cells with EGFP-mRNA. The efficacy was significantly higher than that observed for bPEI 25k. Altogether, this allows us to consider random amphiphilic copolymers as promising candidates for mRNA delivery applications, such as construction of antiviral and anticancer vaccines, and for therapeutic induction of missing proteins expression.

## Figures and Tables

**Figure 1 ijms-23-05363-f001:**
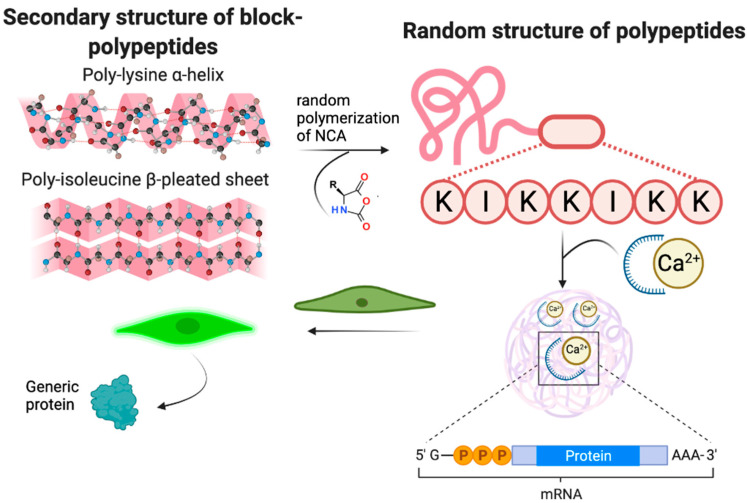
Flexibility of polyplexes based on P(Lys-co-Ile) as compared to PLys (K) and PIle (I), and its possible effect on transfection.

**Figure 2 ijms-23-05363-f002:**
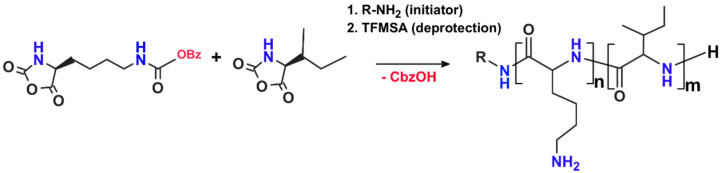
Scheme of the synthesis of P(Lys-co-Ile) via ring-opening polymerization of corresponding NCAs.

**Figure 3 ijms-23-05363-f003:**
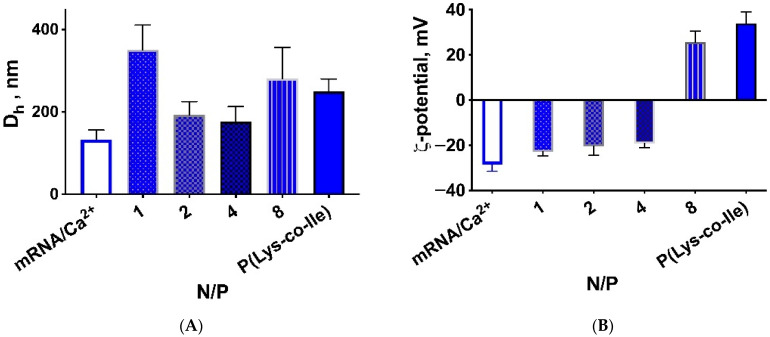
The effect of cationic polypeptide addition to mRNA/Ca^2+^ at different proportions on size (**A**) and ζ-potential (**B**) of formed polyplexes. The presented data were obtained for P(Lys-co-Ile)-3 copolymer. The measurements were performed in ultrapure H_2_O. Results are given as Mean ± SD (*n* = 3).

**Figure 4 ijms-23-05363-f004:**
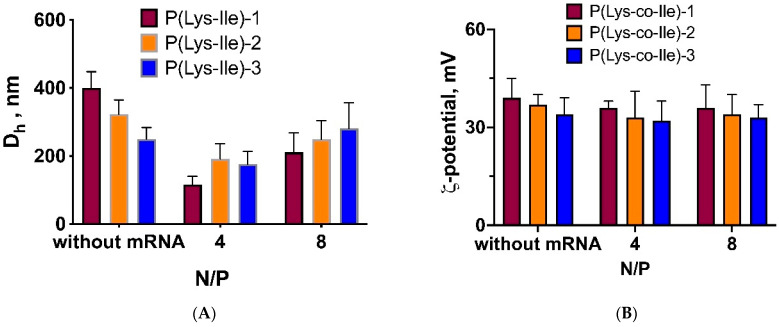
Hydrodynamic diameter (**A**) and ζ-potential (**B**) of cationic polypeptides and polyplexes obtained under the 30 s ultrasonication (10% power from 70 W, 0.21 kJ) during mRNA/Ca^2+^ encapsulation. The measurements were performed in ultrapure H_2_O. Results are given as Mean ± SD (*n* = 3).

**Figure 5 ijms-23-05363-f005:**
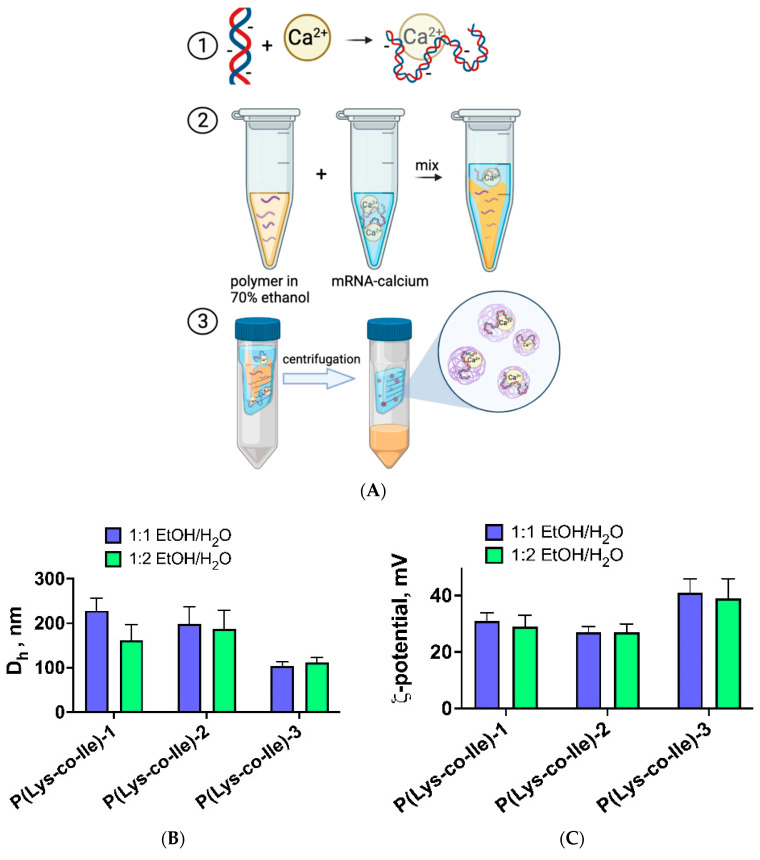
Encapsulation of Mrna via phase inversion: general idea of the method (**A**); size (**B**) and ζ-potential (**C**) of polyplexes obtained at different ethanol/water ratios. N/P ratio was fixed at 4 in these experiments. The measurements were performed in ultrapure H_2_O. Results are given as Mean ± SD (*n* = 3).

**Figure 6 ijms-23-05363-f006:**
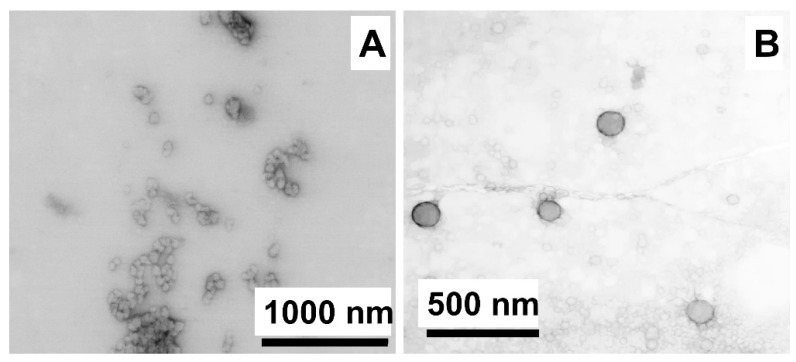
TEM images of p(Lys-co-Ile)-3/Mrna-Ca^2+^ polyplexes obtained by: ultrasonication (**A**) and phase inversion method (**B**).

**Figure 7 ijms-23-05363-f007:**
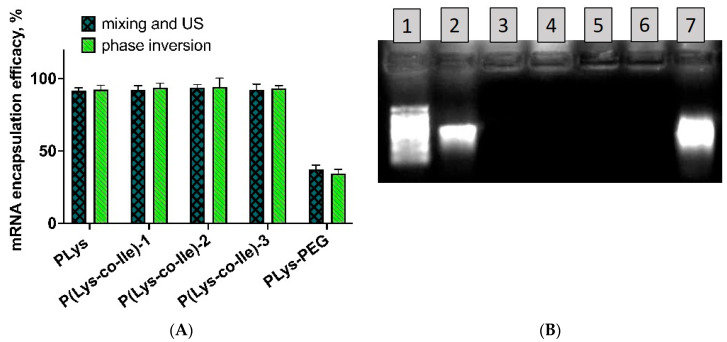
Encapsulation of mRNA: (**A**)—RiboGreen assay-based comparison of mRNA encapsulation efficiency within polyplexes obtained by mixing of components under ultrasound treatment (US) with those obtained by phase inversion. N/P was 4:1 in this experiment. Results are given as Mean±SD (*n* = 3); (**B**)—1% agarose denaturing gel electrophoresis, showing efficient encapsulation of mRNA via phase inversion. P(Lys-co-Ile)-3 was applied in this experiment. Lane 1—Ribo Ruller HighRange; Lane 2—initial mRNA control (600 ng); Lane 3—PLys+mRNA/Ca^2+^ N/P = 4 (~350 ng mRNA); Lane 4—PLys+mRNA/Ca^2+^ N/P = 8 (~350 ng mRNA); Lane 5—P(Lys-co-Ile)-3 +mRNA/Ca^2+^ N/P = 4 (~350 ng mRNA); Lane 6—P(Lys-co-Ile)-3+mRNA/Ca^2+^ N/P = 8 (~350 ng mRNA); Lane 7—mRNA/Ca2+ control (1000 ng).

**Figure 8 ijms-23-05363-f008:**
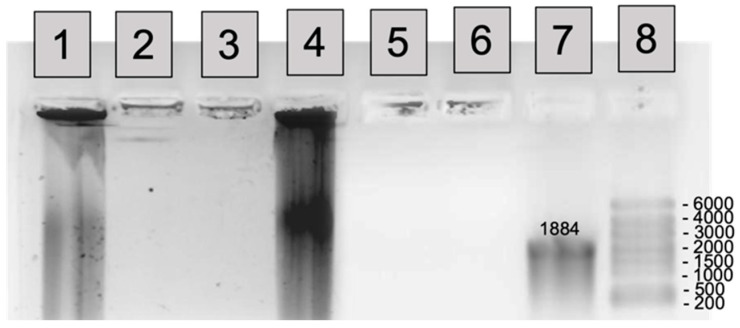
Stability of mRNA within complexes with P(Lys-co-Ile) towards degradation by RNase. 1% agarose denaturing gel electrophoresis at 100 V for 15 min and 120 V for 45 min. Lane 1—PLys+mRNA/Ca^2+^ N/P = 2 + RNase; Lane 2—PLys+mRNA/Ca^2+^ N/P = 4 + RNase; Lane 3—PLys+mRNA/Ca^2+^ N/P = 8 + RNase; Lane 4—P(Lys-co-Ile)-3+mRNA/Ca^2+^ N/P = 2 + RNase; Lane 5—P(Lys-co-Ile)-3+mRNA/Ca^2+^ N/P = 4 + RNase; Lane 6—P(Lys-co-Ile)-3+mRNA/Ca^2+^ N/P = 8 + RNase; Lane 7—initial mRNA control (600 ng); Lane 8—Ribo Ruller HighRange.

**Figure 9 ijms-23-05363-f009:**
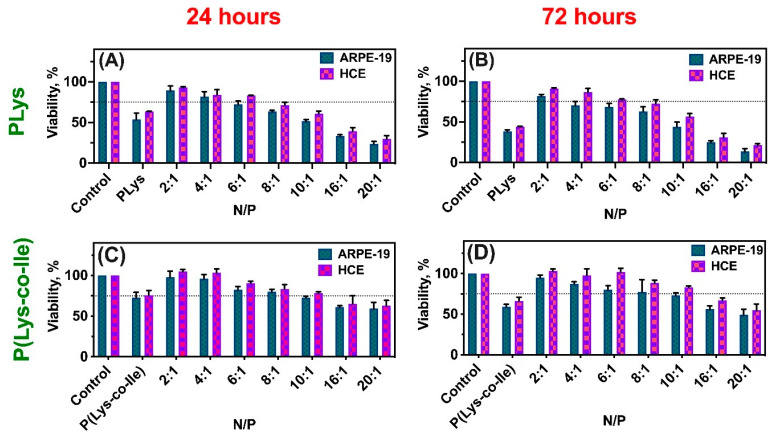
MTT test results: viability of human retinal pigment epithelial cells (ARPE-19) and human corneal epithelial cells (HCE) at different time points: (**A**) PLys and polyplexes after 24 h; (**B**) PLys and polyplexes after 48 h; (**C**) P(Lys-co-Ile)-3 and polyplexes after 24 h; (**D**) P(Lys-co-Ile)-3 and polyplexes after 48 h. Different N/P ratios were tested. Pure PBS, pH 7.4 served as control (100% of viability). Results are given as Mean ± SD (*n* = 3).

**Figure 10 ijms-23-05363-f010:**
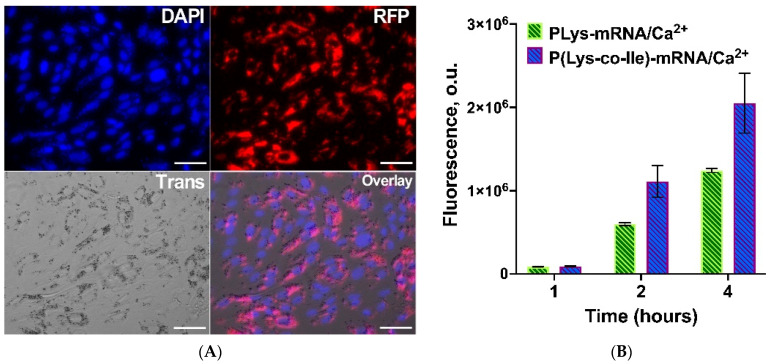
Intracellular penetration studies of P(Lys-co-Ile)-3 +EGFP-mRNA/Ca^2+^ + oligo-dTdA-Cy3 (at N/P 4) particles into the ARPE-19 cells: (**A**) images obtained by fluorescent microscopy after 2 h of co-incubation. Scale bar is 20 µm; (**B**) growth of Cy3 fluorescence signal as measured with Varioscan plate-reader. Results are given as Mean ± SD (*n* = 3).

**Figure 11 ijms-23-05363-f011:**
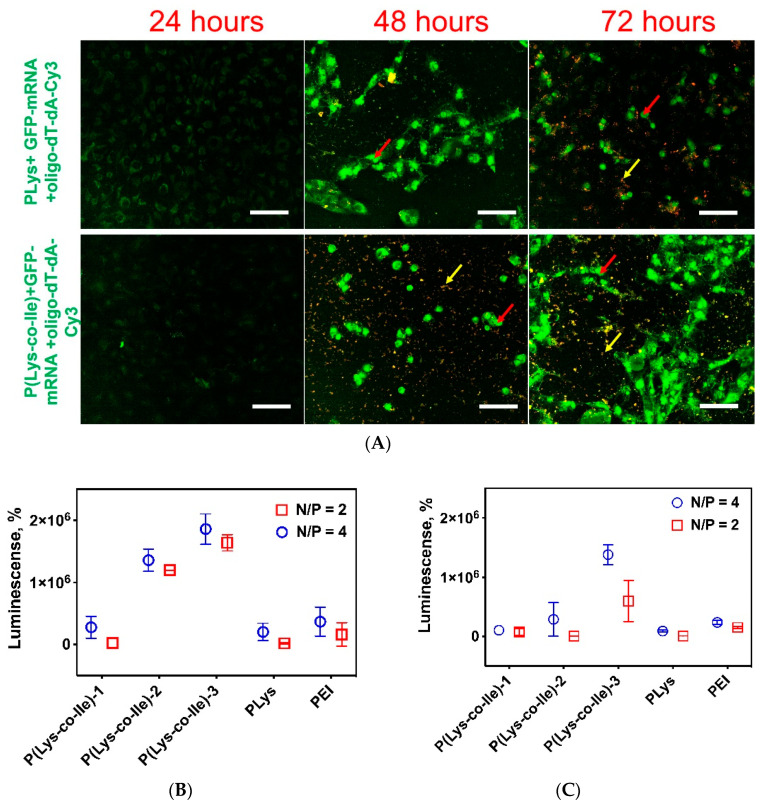
Transfection studies: (**A**) images ARPE-19 cells transfected by EGFP-mRNA- Ca+Cy3-olig-dT-dA at different times of co-incubation. Yellow arrow shows the presence of Cy3 labeled olgonucleotide. Red arrow shows GFP fluorescence. Scale bar is 20 µm; (**B**) results of transfection of ARPE-19 cells with fLuc-mRNA-Ca^2+^ via application of different polypeptide cations with different MW as compared to transfection by bPEI 25k; (**C**) results of transfection of HCE cells with fLuc-mRNA-Ca via different polypeptide polycations under study as compared to transfection by bPEI 25k; of and ARPE-19 using particles. The percentage of luminescence in the tested samples was normalized relative to the luminescence after transfection using bPEI-fLuc-mRNA, which was taken as 100%.

**Table 1 ijms-23-05363-t001:** Composition and molecular-weight characteristics of synthesized polypeptides and characteristics of particles obtained from such copolymers. Lys(Z)/Ile ratio was 80/20 for all samples in this Table. For other ratios please see Appendix A in Appendix A.

Sample	[M]:[I] ^a^	Composition of Copolymers (mol%) (HPLC Analysis ^b^)		Molecular-Wieght Characteristics (SEC Data ^c^)			Particles’ Characteristics (DLS and ELS Data ^d^)		
		Lys	Ile	Mn	Mw	Ɖ	D_H_, nm	PDI	ζ-Potential, mV
PLys	50			7.700	9.700	1.26	−	−	−
P(Lys-co-Ile)-1	50	85.7 ± 1.5	14.3 ± 0.7	6.300	8.100	1.28	400	0.12	+39 ± 6
P(Lys-co-Ile)-2	20	87.0 ± 1.2	13.0 ± 0.5	2.800	3.430	1.22	223	0.19	+37 ± 3
P(Lys-co-Ile)-3	10	84.9 ± 0.9	15.1 ± 0.4	1.300	1.770	1.36	189	0.18	+34 ± 5

^a^ monomers to initiator ratio. ^b^ quantitative HPLC analysis of amino acids was carried after total acidic hydrolysis of copolymers to free amino acids. ^c^ data were obtained for Z-protected polymers with application of PMMA standards. ^d^ data were obtained for the self-assembled polymer particles after removal of Z-protection. Particles were obtained by ultrasonication during 30 s. Size and ζ-potential were measured in ultrapure water.

## Data Availability

The authors confirm that the data supporting the findings of this study are available within the article and its Appendix A.

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
