# Peer review of "Random Copolymers of Lysine and Isoleucine for Efficient mRNA Delivery"

_ijms, 2022, doi:10.3390/ijms23105363_

Round 1
Reviewer 1 Report
This manuscript uses co-oligomers of lysine (K) and isoleucine to generate mRNA-calcium salt nanoparticles. > 95% Binding was only achieved with 80% K, and lowering the K to 70% led to much less binding. The effect of the chain length was investigated, but appears to be minor. It required quite a large excess of polycation (4 < equalisation < 8) to coat the NP with simple mixing. They then investigated sonication, and acquired small, zeta-positive nanoparticles at reasonable N:P ratios.
Whilst effective, sonication may be avoided by solvent inversion, which was more effective at producing spherical, unaggregated nanoparticles. All three polymers and all three techniques gave similar encapsulation. Only the 10-mer (K8I2) was investigated further. It transfected better than PLL, but the PLL was DP = 50, and was being compared to DP = 10 cooligomer. One wonders why a DP10 PLL wasn’t used, as a direct comparison, but this is a minor point.
- 1 has no calcium chloride in it.
- 2 – the omega terminus should be a proton
- Table 1 – better to use letters for annotations – I spent some time wondering why DH cubed was there
- There is a j value in the NMR, which should be paired.
- The PEI used as a control is 25 kDa branched PEI. The properties of PEI vary a lot with MW and conformation. Use a more explicit name, such as bPEI-25k. In fig. 1 PEI of Mn 5,000 is used. Is this a different PEI?
- The note on Fig S3 – yes, sort out the names
- Check English
- "spherical"vice spheric
- "Cell culture"
- The structure and purity of NCAs were testified by - confirmed but testified
Author Response
Authors greatly appreciate the interest, attention and comments of reviewers, as well as their efforts for increasing the manuscript quality. The authors tried to do their best to improve the Manuscript according to the reviewers report.
Our corrections are marked with red text. In addition, the references to the reviewers comments are made at the corrections mode of MS Word.
Our replies to the comments are below.
Comment 1: has no calcium chloride in it.
Reply: Thank you for your valuable comment. We did our best to improve the Scheme. Please see the corrected Figure within the new version of MS (Page 4, line 136).
Comment 2: the omega terminus should be a proton
Reply: According to advice of respected reviewer we have corrected the Figure 2.
Comment 3: Table 1 – better to use letters for annotations – I spent some time wondering why DH cubed was there.
Reply: We agree with this comment and have made the corresponding corrections in Tables 1 and S1.
Comment 4: There is a j value in the NMR, which should be paired.
Reply: We failed to determine j values in polymer NMRs due to relaxation reasons. For this reason, we cannot add them to MS.
Comment 5: The PEI used as a control is 25 kDa branched PEI. The properties of PEI vary a lot with MW and conformation. Use a more explicit name, such as bPEI-25k. In fig. 1 PEI of Mn 5,000 is used. Is this a different PEI?
Reply: Thank you for reading the manuscript carefully. We improved the abbreviations according to your advice. We are sorry for the mistake. There was the same bPEI 25k in all experiments.
Comment 6: The note on Fig S3 – yes, sort out the names
- Check English
- "spherical"vice spheric
- "Cell culture"
- The structure and purity of NCAs were testified by - confirmed but testified
Reply: Thank you for your careful reading of this manuscript and your valuable comments, we carefully checked the manuscript language and made significant improvements.

Reviewer 2 Report
ijms-1697935 International Journal of Molecular Science 2022
The authors of “Random Copolymers of Lysine and Isoleucine for Efficient 2 mRNA Delivery” describe the use of random poly(lysine-co-isoleucine) polypeptides for gene delivery purposes.
The development of alternative shuttles for DNA and RNA encapsulation is an interesting issue and the manuscript describes interesting results. The work is quite complete, however, a certain disorder in results presentation and the polyplexes characterization showed to make the work difficult to follow and understand. Moreover, the reviewer suggests better clarifying the composition of the analysed polymers, if indeed they are copolymers, are they main blocks with amphiphilic properties?
Accordingly, the reviewer suggests its publication with major revisions.
Referee remarks/questions:
- The table and figure legends must be complete with all the information necessary to understand them.
- What’s the meaning of [M]:[I] ?
- What’s the pH of the polymers in water?
- In line 158 the reviewer doesn’t understand the meaning of the sentence “P(Lys-co-Ile) copolymers with molar contents of n and m (see Fig. 2) equal 60/40 and 70/30”. Do the authors really refer to figure 2?
- Table S1, SEC data of 80:20 [Lys]:[Ile] that are finally chosen for further experiments are omitted, why?
- Figure 5 A the scheme proposed is not complete to describe the methods. It must be integrated.
- Regarding the encapsulation of mRNA Figure 7 B, the authors have to solve better the agarose gel electrophoresis. It is impossible to distinguish the molecular RNA marker and the size of mRNA. The mRNA seems to be smaller than the shown in figure 8.
- line 280 The % of agarose gel doesn’t correspond to the MM section.
Author Response
Authors greatly appreciate the interest, attention and comments of reviewers, as well as their efforts for increasing the manuscript quality. The authors tried to do their best to improve the Manuscript according to the reviewers report.
Our corrections are marked with red text. In addition, the references to the reviewers comments are made at the corrections mode of MS Word.
Our replies to the comments are below.
Comment 1: the reviewer suggests better clarifying the composition of the analyzed polymers, if indeed they are copolymers.
Reply: Thank you very much for this valuable comment. We did our best to improve the MS in this matter. We have added HPLC data on copolymers composition to Table 1 and Table S1. We have also made corresponding corrections in the text.
Page 4, line 157: “The amino acid composition of copolymers was determined via quantitative HPLC analysis of amino acids obtained after total acidic hydrolysis of copolymers (Fig. S1). It was found that using of the Lys/Ile initial ratio equal to 80/20 (mol/mol) resulted in the formation of copolymers consisting of 85-87 mol% of Lys and 13-15 mol% of Ile (Table 1). This is caused by greater activity of Lys NCA in ring-opening polymerization as compared to Ile NCA.”
Page 16, line 540: “Composition of copolymers was determined via quantitative HPLC analysis of free amino acids obtained after total acidic hydrolysis of copolymers with the use of protocol developed earlier [33]. An example of chromatogram as well as details on chromatographic conditions and calibration plots can be found in Supplementary Materials (Fig. S1).”
Comment 2: The table and figure legends must be complete with all the information necessary to understand them.
Reply: We did our best to improve the figure legends contents. Please, look through MS for made corrections.
Comment 3: What’s the meaning of [M]:[I] ?
Reply: [M]:[I] refers to the monomer to initiator ratio, which is very important factor for polymerization process. The corresponding footnote (Page 5, line 166) was added to the Table 1 in order to clarify this point.
Comment 4: What’s the pH of the polymers in water?
Reply: The authors doubt they got the question right. The pH of the water changes upon addition of P(Lys-co-Ile), which is polyelectrolyte. After addition of 1 mg of polymer the pH of MQ water turns from about 7.0 to 8.2. After addition of 0.1 mg of the copolymer, the pH of water solution is about 7.3. However, only 2-20 µg of polymer (depending on N/P) were used for transfection studies. Thus, we assume that effect of polymers on pH is negligible. It is also necessary to note, that transfection is performed in buffered systems. Thus, there is no valuable change of pH as a result of copolymer addition.
If the question is about the degree of protonation of polymers in water, we can refer the respected Reviewer back to presented zeta-potential values (please, see Figs 3 and 4).
Comment 5: In line 158 the reviewer doesn’t understand the meaning of the sentence “P(Lys-co-Ile) copolymers with molar contents of n and m (see Fig. 2) equal 60/40 and 70/30”. Do the authors really refer to figure 2?
Reply: Here we wanted to show what are n and m, which are shown on the Fig. 2. However, according to your comment we decided to revise the sentence.
Page 5, line 182: The text was corrected as follows: “P(Lys-co-Ile) copolymers with Lys/Ile molar ratios in polymerization mixture equal 60/40 and 70/30 gave quite large particles and poorly bind mRNA (see Table S1).”
Comment 6: Table S1, SEC data of 80:20 [Lys]:[Ile] that are finally chosen for further experiments are omitted, why?
Reply: We apologize for this inaccuracy. In the revised version, we have provided the SEC data for all copolymers in Table S1.
Comment 7: Figure 5 A the scheme proposed is not complete to describe the methods. It must be integrated.
Reply: The scheme on Figure 5A has been corrected. We tried to insert all stages of polyplexes preparation via phase inversion approach.
Comment 8: Regarding the encapsulation of mRNA Figure 7 B, the authors have to solve better the agarose gel electrophoresis. It is impossible to distinguish the molecular RNA marker and the size of mRNA. The mRNA seems to be smaller than the shown in figure 8.
Reply: We apologize for the made mistake and greatly thank the reviewer for this comment! We have carefully checked and readjusted the relevant contents. In previous variant of our MS in Figure 8 we presented DNA ladder and DNA stability test. In the revised version of the manuscript, we have moved this data to Supplementary materials. In turn, the correct mRNA denaturating electrophoresis was inserted as Figure 8. Ribo Ruller High Range marker (see Fig. 8) was used there. The corrected figure also shows that encapsulation of the mRNA inside the polymer nanoparticles at N:P ratios starting from 4:1 and higher led to prevention of mRNA degradation by RNase.
Page 10, line 309. The text was corrected according to the renewed Figure 8: “The degradation and stability of mRNA were detected by 1.0 % agarose denaturing gel electrophoresis (Fig. 8). Comparison of lanes 1 and 4 clearly shows that unprotected at low N/P 2 mRNA can be degraded by RNase. Complexation of mRNA within polyplexes based on PLys and P(Lys-co-Ile) at N/P 4 and N/P 8 resulted in the stabilization of mRNA towards degradation by RNase (Lanes 2,3 and 5,6 respectively).
In addition, the ability of polymers to stabilize the DNA was proved (Fig. S4, Supplementary materials).”
Comment 9: line 280 The % of agarose gel doesn’t correspond to the MM section.
Reply: Please see our reply to the previous comment. The agarose electrophoresis figure was replaced with a correct one.

Reviewer 3 Report
The paper is well written and interesting but would benefit from additional details in methods and inclusion of appropriate controls.
Luciferase activity should be normalized to protein concentration to consider cytotoxicity (RLU/mg of proteins).
A commercial standard such as Lipofectamine Messenger Max or Jet Messenger needs to be included in transfectio experiments.
Authors should replace "GFP" by "EGFP".
EGFP mRNA expression should be quantitated by flow cytometry. Authors should explain why they detect an expression peak at 48h whereas most if not all studies on mRNA delivery exhibit a peak of mRNA expression at 10-24h.
Authos need to discuss Poly Ion Complexes and LNP for mRNA delivery in the Introduction.
Authors should indicate whether polyplexes were incubated in serum-containing media or just DMEM (fig. S1).
Authors need to justify the requirement of calcium for mRNA complexation as all other studies on mRNA delivery systems (polyplexes, Poly Ion Complexes, LNP) do not need calcium for mRNA complexation). Moreover, Authors need to discuss potential harmfull effects of perturbating calcium homeostasis.
mRNA gele electrophoresis or Agilent results need to be provided to confirm the quality of the mRNA used.
For ultrasonication, the acoustic energy used (Joules) should be indicated.
Authors need to perform cellular uptake experiments with labeled mRNA not oligonucleotides as cell permeation and intracellular mobility could be size-dependent.
Authors need to determine the best N/P ratio for mRNA delivery rathe than assuming it is the same as pDNA.
Authors need to explain the y axis legends of Fig 11A-11B: what do percentages mean? Do they mean flod luciferase expression with 1 for bPEI?
Authors should explain what "vector antiviral vaccines" means.
Author Response
Authors greatly appreciate the interest, attention and comments of reviewers, as well as their efforts for increasing the manuscript quality. The authors tried to do their best to improve the Manuscript according to the reviewers report.
Our corrections are marked with red text. In addition, the references to the reviewers comments are made at the corrections mode of MS Word.
The paper is well written and interesting but would benefit from additional details in methods and inclusion of appropriate controls.
Comment 1: Luciferase activity should be normalized to protein concentration to consider cytotoxicity (RLU/mg of proteins).
Reply: Thank you very much for the valuable comment, as well as for important guidance for our further work. It is impossible for us to repeat this experiment now, however, we will definitely use this type of normalization in future. Only we can comment here, that during this experiment we have observed cells viability microscopically and have detected their constant viability and confluence during experiment.
Also, cytocompatibility of the nanoparticles was assessed via detection of mitochondrial activity of the cells. Summarizing data on cytotoxicity and cell transfection, including visual fluorescence observation of the cells transfected, we can conclude that polymer nanoparticles at low N/P 4 have quite good transfection efficiency and cytocompatibility as well.
Comment 2: A commercial standard such as Lipofectamine Messenger Max or Jet Messenger needs to be included in transfection experiments.
Reply: Thank you for your valuable advice! In fact, we used Lipofectamine 2000 Transfection agent (Invitrogen) to prepare a lipoplexes with mRNA-EGFP in additional flow cytometry experiment (see reply to comment 4). However, in our experiments on P(Lys-co-Ile) transfection we intentionally have used branched PEI as cationic polymer with well-known transfection efficacy. Thus, bPEI in our opinion could serve here as appropriate model for comparative studies with different polyplexes.
Comment 3: Authors should replace "GFP" by "EGFP".
Reply: Done. Corrections were made through all the text.
Comment 4: EGFP mRNA expression should be quantitated by flow cytometry. Authors should explain why they detect an expression peak at 48h whereas most if not all studies on mRNA delivery exhibit a peak of mRNA expression at 10-24h.
Reply: According to the advice of the Reviewer, we have added the data on the efficiency of copolymers to deliver the EGFP-mRNA payload into K562 cancer cell line. The experiment was performed by flow cytometry. We did not include this information in the main body of the firstly submitted paper, since another cell line was used in that experiment. Here we used naked EGFP-mRNA (Fig.S5) and Lipofectamine 2000 transfection agent as controls. As you mentioned, lipofectamine represented higher transfection efficiency after 24 hours of post-transfection rather than 48 hours. However, cells transfected by polymer nanoparticles represented higher GFP signal after 48 hours that was additionally proved by fluorescence microscopy of GFP-producing cells (Fig.11). We sincerely hope the revised manuscript meets your approval.
Page 12, Line 382, the following text was added to the MS: “The transfection efficacy of K562 immortalized myelogenous leukemia cell line with EGFP-mRNA was tested via flow cytometry (Fig. S6). In this experiment naked EGFP-mRNA and this complexed with Lipofectamine 2000 transfection agent were used as controls, while P(Lys-co-Ile) polyplex with EGFP/mRNA/Ca2+ was the sample under study. One can observe that transfection with lipofectamine is the most after 24 hours, while for P(Lys-co-Ile) the maximum transfection was observed only after 48 hours. This observation could be explained by greater size of polyplex particles than those of lipoplexes. It is known that smaller particles better penetrate cellular interior and better act as transfection agents [48-50].”
Comment 5: Authors need to discuss Poly Ion Complexes and LNP for mRNA delivery in the Introduction.
Reply: The information on lipoplexes and LNPs was added to the Introduction. The reference on polyion complexes was also added.
Page 2, Lines 54-82:
“There are three major types of nanoplatforms applied for non-viral genetic constructions intracellular delivery: lipoplexes, lipid nanoparticles and polyplexes [13]. First ones are composed of positively charged lipids, which form self-organized spherical vesicles due to presence of distinct hydrophilic and hydrophobic parts within their molecules, and could encapsulate nucleic acids (NA) into their inner aqueous phase [14]. Lipoplexes rep-resent quite efficient class of transfection agents, which efficiently penetrate cellular interior and escape from endosomes [15–17]. Despite good efficacy in vitro lipoplexes possess some limitations for in vivo applications, such as uncertain stability in serum [18] and toxicity [19].
These features of lipoplexes have turned the interest to lipid nanoparticles (LNPs) as NA delivery vehicles [20,21]. The morphology of LNPs differs from traditional liposome bilayer, and characterized by inverted micelle formed by cationic/ionizable lipids around the encapsulated NA molecules [22]. LNPs are more stable and quite versatile systems, which could serve as efficient NA delivery nanoplatforms in different applications [23]. However, further development of LNPs involves the possibility of combining LNPs with polycations in order to increase their intracellular penetration, endosomal escape and transfection efficacy [21].
Polycations represent an important type of non-viral vectors for delivery of genetic constructions [24,25]. During interaction of polycations with NA the condensation of NA into so called polyplexes (or polyion complexes) occurs [26]. This process is quite simple and reproducible, which makes it very attractive for future clinical applications. The ob-tained polyplexes are nanoparticles with high density leading for easy cell internalization and enhanced protection from enzymatic degradation [25]. The study of polycations ap-plication for stabilization of NA both in vitro and in vivo, as well as for promotion of nu-cleic acids intracellular penetration is vast and emerging. However, clinical application of polycations is still limited, mainly due to the low efficacy of such vectors. In order to give polycations a chance to serve as effective NA delivery vectors the study of different structural peculiarities of such polycations on the efficacy of NA binding, stabilization and transfection is of great importance.”
Comment 6: Authors should indicate whether polyplexes were incubated in serum-containing media or just DMEM (fig. S1).
Reply: Thank you for reading the manuscript carefully and providing this comment. We used serum-free OptiMEM for that experiment, we have accordingly improved the information (Page 16, line 554; Supplementary materials, Page 2, line 48).
Comment 7: Authors need to justify the requirement of calcium for mRNA complexation as all other studies on mRNA delivery systems (polyplexes, Poly Ion Complexes, LNP) do not need calcium for mRNA complexation). Moreover, Authors need to discuss potential harmfull effects of perturbating calcium homeostasis.
Reply: Thank you for your interesting and valuable question!
By using calcium complexation, the particle size and mRNA complexation were significantly improved. However, no potential toxic effects were observed due to the low concentration of calcium chloride used.
Moreover, from our experience working with amino-acid oligomers and mRNA, direct addition of NA usually results in greater nanoparticles size starting from 440 nm and higher. Using calcium chloride at a very low concentration compared to normal concentration of Ca in cell medium can be accepted as safe method for mRNA pre-complexation.
As you mentioned, Ca2+ is a ubiquitous intracellular messenger that controls diverse cellular functions but can become toxic and cause cell death. Intracellular free Ca2+ concentration widely varies depending on its location. The cytoplasmic [Ca2+] ([Ca2+]c) under resting conditions is ∼10-7M, 104 times lower than [Ca2+] in the extracellular millieu (∼10-3M). Inside the cell, Ca2+ levels in the nuclear matrix ([Ca2+]n) and in the mitochondrial matrix ([Ca2+]mt) are similar to that in the cytoplasm. However, other intracellular organelles, known as Ca2+ stores, can accumulate Ca2+ and maintain a higher [Ca2+] than the cytoplasm (1-5×10-4M) [Bagur, R., & Hajnóczky, G. (2017). Intracellular Ca2+ sensing: its role in calcium homeostasis and signaling. Molecular cell, 66(6), 780-788].
In our study, we used calcium at a concentration of 1.4 ´ 10-7 M to provide additional complexation and further successful encapsulation of mRNA-Calcium complexes into polymer nanoparticles with a very low-dispersity and nano-size (∼200 nm and PDI 0.2).
We additionally revised the manuscript and added following information.
Page 6, line 204: “It should be noted that in all cases mRNA was initially condensed by addition of Ca2+ ions. By using calcium complexation, the particle size and mRNA complexation were significantly improved. Notable, that no harmful effects of such complexation on cellular calcium homeostasis [41] were observed due to the low concentration of calcium ions used.”
Page 17, line 589: “Calcium chloride was overall used at a concentration of 1.4 ´ 10-7 M established as safe concentration for intra- and extracellular environment [41]. ”
Comment 8: mRNA gel electrophoresis or Agilent results need to be provided to confirm the quality of the mRNA used.
Reply: 1 % agarose denaturing gel electrophoresis, showing mRNA stability obtained from different vials after synthesis and storage was inserted to Supplementary materials (see Fig. S7). The corresponding corrections to the text were also provided.
Page 17, line 583: “The stability of Luc mRNA was testified before application by 1 % agarose denaturing gel electrophoresis (Fig. S7, Supplementary materials).”
Comment 9: For ultrasonication, the acoustic energy used (Joules) should be indicated.
Reply: Bandelin Sonopuls with Ultrasonic generator GM 2070.2 was used. Its maximum output is 70 W, but we used only 10 % for 30 seconds. Thus, the acoustic energy, which act on the sample is 210 J (0.21 kJ). We have added this information to the experimental part (see Page 16, line 549).
Comment 10: Authors need to perform cellular uptake experiments with labeled mRNA not oligonucleotides as cell permeation and intracellular mobility could be size-dependent.
Reply: We agree with reviewer that cellular uptake is size dependent. However, labeled mRNA was not available for us. For these reasons we have performed cellular uptake experiments with application of polyplexes loaded with both mRNA and labeled oligonucleotides. The size of these particles was very close to the size of particles loaded with just mRNA.
Page 11, line 354. The text was corrected as follows: “In order to detect the intracellular penetration of polyplexes Cy3 labeled oligonucleotide duplex, namely oligo-dT-dA (Cy3-oligo-dT-dA), was included into composition of these polyplexes. The hydrodynamic diameter of P(Lys-co-Ile)/EGFP-mRNA/ Cy3-oligo-dT-dA particles (162±34 nm) was similar to those of just P(Lys-co-Ile)/EGFP-mRNA (155±41 nm).”
The corresponding corrections were also made in the captions of the figures.
Comment 11: Authors need to determine the best N/P ratio for mRNA delivery rather than assuming it is the same as pDNA.
Reply: We completely agree with reviewer that N/P ratios for pDNA and mRNA transfection could be different. Moreover, we have tested transfection with application of mRNA at two different N/P ratios, but not at only one, which was found optimal. Thus, we deleted the confusing words from the discussion of results obtained with pDNA transfection.
Page 12, line 377. The text under discussion now appears as following: “It was observed that transfection of pDNA with application of both PLys and P(Lys-co-Ile) is less effective than that provided by bPEI 25k. At the same time P(Lys-co-Ile) showed better efficacy towards transfection of pDNA than just PLys. One can observe that most efficient transfection with application of P(Lys-co-Ile) was observed at N/P 4.”
Comment 12: Authors need to explain the y axis legends of Fig 11A-11B: what do percentages mean? Do they mean flod luciferase expression with 1 for bPEI?
Reply: Thank you for careful reading of manuscript. We are sorry for vague explanation. Here, the percentage was compared to fly luciferase expression provided by bPEI, which was taken as 100 %.
Page 19, line 698. The following text was added to the experimental part: “The percentage of luminescence in the tested samples was normalized relative to the luminescence after transfection using bPEI-fLuc-mRNA, which was taken as 100%.”
Comment 13: Authors should explain what "vector antiviral vaccines" means.
Reply: Sorry for this mistake. The sentence was corrected – word “vector” was deleted.
Page 19-20, lines 715-718. The text corrected as follows: “Altogether, this allows us to consider random amphiphilic copolymers as promising candidates for mRNA delivery applications, such as construction of antiviral and anticancer vaccines, and for therapeutic induction of missing proteins expression.”

Round 2
Reviewer 2 Report
The reviewer agrees with the authors' changes / corrections. Some additional minor corrections are required for publication.
- The authors should substitute MQ H2O, ( if they referred to Milli-Q® that is a trademark) with ultrapure water.
- Figures 7, 8, S4 and S7. To help the reader, the authors must add "Line" with the corresponding number (Line 2: Line 3 etc.), identifying them, for all the lines of the gels.
Author Response
Authors greatly appreciate reviewers’ time and efforts for increasing the manuscript quality! Thank you very much!
Comment 1: The authors should substitute MQ H2O, ( if they referred to Milli-Q® that is a trademark) with ultrapure water.
Reply: Thank you for this comment. We have changed “MQ” to “ultrapure” in all figure captions and text.
Comment 2: Figures 7, 8, S4 and S7. To help the reader, the authors must add "Line" with the corresponding number (Line 2: Line 3 etc.), identifying them, for all the lines of the gels.
Reply: Thank you for this comment. We agree that this will increase the understanding of the paper contents. The corresponding corrections were made in figures 7, 8, S4 and S7 captions.
Reviewer 3 Report
Authors adressed all comments so I recommend acceptance of the revised paper.
Author Response
Comment 1: Authors addressed all comments so I recommend acceptance of the revised paper.
Reply: Authors greatly appreciate reviewers’ time and efforts for increasing the manuscript quality! Thank you very much!